# Risk of miscarriage in women with chronic diseases in Norway: A registry linkage study

Maria C. Magnus[1,2,3]*, Nils-Halvdan Morken[1,4,5], Knut-Arne Wensaas[6], Allen J. Wilcox[7], Siri E. Håberg[1]

**1** Centre for Fertility and Health, Norwegian Institute of Public Health, Oslo, Norway, **2** MRC Integrative Epidemiology Unit at the University of Bristol, Bristol, United Kingdom, **3** Population Health Sciences, Bristol Medical School, Bristol, United Kingdom, **4** Department of Clinical Science, University of Bergen, Bergen, Norway, **5** Department of Obstetrics and Gynecology, Haukeland University Hospital, Bergen, Norway, **6** Research Unit for General Practice, NORCE Norwegian Research Centre, Bergen, Norway, **7** Epidemiology Branch, National Institute of Environmental Health Sciences, National Institutes of Health, Durham, North Carolina, United States of America

* Maria.Christine.Magnus@fhi.no

## Abstract

### Background

Increased risk of miscarriage has been reported for women with specific chronic health conditions. A broader investigation of chronic diseases and miscarriage risk may uncover patterns across categories of illness. The objective of this study was to study the risk of miscarriage according to various preexisting chronic diseases.

### Methods and findings

We conducted a registry-based study. Registered pregnancies ($n$ = 593,009) in Norway between 2010 and 2016 were identified through 3 national health registries (birth register, general practitioner data, and patient registries). Six broad categories of illness were identified, comprising 25 chronic diseases defined by diagnostic codes used in general practitioner and patient registries. We required that the diseases were diagnosed before the pregnancy of interest. Miscarriage risk according to underlying chronic diseases was estimated as odds ratios (ORs) using generalized estimating equations adjusting for woman's age. The mean age of women at the start of pregnancy was 29.7 years (SD 5.6 years). We observed an increased risk of miscarriage among women with cardiometabolic diseases (OR 1.25, 95% CI 1.20 to 1.31; $p$-value <0.001). Within this category, risks were elevated for all conditions: atherosclerosis (2.22; 1.42 to 3.49; $p$-value <0.001), hypertensive disorders (1.19; 1.13 to 1.26; $p$-value <0.001), and type 2 diabetes (1.38; 1.26 to 1.51; $p$-value <0.001). Among other categories of disease, risks were elevated for hypoparathyroidism (2.58; 1.35 to 4.92; $p$-value 0.004), Cushing syndrome (1.97; 1.06 to 3.65; $p$-value 0.03), Crohn's disease (OR 1.31; 95% CI: 1.18 to 1.45; $p$-value 0.001), and endometriosis (1.22; 1.15 to 1.29; $p$-value <0.001). Findings were largely unchanged after mutual adjustment. Limitations of this study include our inability to adjust for measures of socioeconomic position or lifestyle characteristics, in addition to the rareness of some of the conditions providing limited power.

**Data Availability Statement:** Study data are available on application via helsedata.no, subject to the necessary ethics approvals.

**Funding:** This research was supported by the Research Council of Norway through its Centres of

Excellence funding scheme, project number 262700 (SEH, MCM, AJW). The work was also supported by the Intramural Program of the National Institute of Environmental Health Sciences, NIH (AJW). MCM works at the Medical Research Council (MRC) Integrative Epidemiology Unit at the University of Bristol which receives infrastructure funding from the UK MRC (MC_UU_00011/6). The funders had no role in study design, data collection and analysis, decision to publish, or preparation of the manuscript.

**Competing interests:** The authors have declared that no competing interests exist.

**Abbreviations:** ICD, International Classification of Diseases; ICPC-2, International Classification of Primary Care; OR, odds ratio; PCOS, polycystic ovary syndrome; RECORD, REporting of studies Conducted using Observational Routinely-collected Data; STROBE, Strengthening the Reporting of Observational Studies in Epidemiology.

## Conclusions

In this registry study, we found that, although risk of miscarriage was largely unaffected by maternal chronic diseases, risk of miscarriage was associated with conditions related to cardiometabolic health. This finding is consistent with emerging evidence linking cardiovascular risk factors to pregnancy complications.

## Author summary

### Why was this study done?

- Scattered findings indicate that women with different chronic disease have an increased risk of miscarriage.

- To our knowledge, none of the existing studies have evaluated a broader range of chronic diseases to further evaluate the role of common as opposed to independent biological mechanisms.

### What did the researchers do and find?

- With linked data from Norwegian national registries (including 593,009 pregnancies), we examined the risk of miscarriage according to 25 different chronic diseases diagnosed prior to pregnancy.

- Although risk of miscarriage was largely unaffected by maternal chronic diseases, risk of miscarriage was associated with conditions related to cardiometabolic health.

- Other conditions associated with risk of miscarriage included hypoparathyroidism, Cushing syndrome, Crohn's disease, and endometriosis.

### What do these findings mean?

- This finding is consistent with emerging evidence linking cardiovascular risk factors to pregnancy complications.

- Further investigation is necessary to better understand the underlying mechanisms resulting in an increased risk of miscarriage across a range of chronic diseases.

## Introduction

Miscarriage occurs in 12% to 15% of recognized pregnancies [1–4]. Although the underlying cause of most miscarriages is unknown, they presumably result from a complex interplay between parental age, genetic, hormonal, metabolic, immunological, and environmental factors [5,6]. Genetic factors, including parental chromosomal rearrangements and abnormal embryonic genotypes or karyotypes, are found in more than half of recurrent miscarriages [6].

Various studies of specific illnesses have suggested an association with risk of miscarriage. These include systemic lupus erythematosus [7], type 1 diabetes [8,9], celiac disease [10,11],

asthma/allergies [12], thyroid disorders [13–16], Addison disease [17,18], type 2 diabetes [9,19,20], parathyroid disorders [21,22], Cushing syndrome [23], migraine [24], and epilepsy [25,26]. These scattered findings suggest that immunological, metabolic, or endocrinological mechanisms might be linked to miscarriage risk, any of which is biologically plausible since all play a role in development and function of the placenta [27–30]. However, most of these studies were based on relatively small samples, and most are retrospective and therefore susceptible to recall bias.

The Norwegian health registries contain comprehensive registration of all recognized pregnancies resulting in contact with healthcare services. Our objective was to study the risk of miscarriage in relation to women's chronic illness present prior to pregnancy. We examined the risk of miscarriage across a broad range of illnesses, including autoimmune, cardiometabolic, endocrinological, neurological, allergic, and reproductive.

## Methods

This study is reported as per the Strengthening the Reporting of Observational Studies in Epidemiology (STROBE) and REporting of studies Conducted using Observational Routinely-collected Data (RECORD) guidelines (S1 RECORD Checklist). The study was conducted according to a prespecified analysis plan (S1 Analysis Plan). The only deviation from this plan was that we not only evaluated the risk of miscarriage according to the individual chronic diseases but also according to subgroups which might share underlying disease pathways.

### Study population and data sources

We conducted a registry-based study using Norwegian national registries. The study population consisted of all registered pregnancies in Norway with an estimated date of conception between January 1, 2010 and December 31, 2016. Information on pregnancies came from 3 national health registries: the Medical Birth Registry of Norway, the Norwegian Patient Registry, and the general practitioner database. Hospital discharges in the patient registry are coded according to International Classification of Diseases (ICD) version 10, while the general practitioner database is coded according to the International Classification of Primary Care (ICPC-2) [31]. The birth registry includes information on all pregnancies ending at 12 gestational weeks or later (live births, stillbirths, late miscarriages, and late induced abortions). Miscarriages before 12 weeks were captured by registrations in the patient registry (with information on all contact with specialist healthcare services) and by the general practitioner database (with information on all contact with primary healthcare services). We used information from the patient registry to account for induced abortions. Ectopic pregnancies were excluded from the study. We linked information from the 3 health registries by using the unique personal identification numbers given to all Norwegian citizens. A description of the data sources and linkage procedures is provided in the Supplement Methods (S1 Text). The analysis was based on de-identified data. The study was approved and participant consent was waived by the Regional Committee for Medical and Health Research Ethics South-East, reference number 2014/404.

### Pregnancy outcomes and identification of unique pregnancies

We identified live births and fetal deaths after 12 gestational weeks through the birth registry. We defined stillbirth as a fetal death at 20 gestational weeks or later, or death of a fetus with a birthweight of 500 grams or more. Miscarriage were defined as fetal deaths before 20 gestational weeks where the fetus also had a birthweight less than 500 grams. In addition, we used the patient registry to capture miscarriages before 12 completed gestational weeks using the following ICD-10 codes: hydatidiform mole (O01); blighted ovum and nonhydatidiform mole

(O02.0); missed abortion (O02.1); other specified abnormal products of conception (O02.8); abnormal product of conception, unspecified (O02.9); spontaneous abortion (O03); and threatened abortion (O20.0). These are the diagnostic codes we previously used to identify miscarriages in the patient registry [32]. The following ICD-10 codes were used to define induced abortions in the patient registry: medical abortion (O04), other abortion (O05), and unspecified abortion (O06).

The present study expands our earlier definition of miscarriage by including events reported in the general practitioner database [32]. Registrations of bleeding in pregnancy (ICPC-2 code W03) or spontaneous abortion (ICPC-2 code W82) in the general practitioner database were included as miscarriages if there was no subsequent registration of delivery or birth.

Pregnancies in the patient and general practitioner databases are not registered with unique pregnancy IDs, which means that follow-up visits for the same miscarriage produce multiple registrations. We took the following steps to define unique pregnancies. First, we required a minimum of 6 weeks (42 days) between 2 successive records of miscarriage in the patient registry, and a minimum of 3 months (90 days) between 2 successive records of miscarriage in the general practitioner database, to consider them distinct pregnancies. These cutoffs were chosen by inspecting the distribution of the time between registrations in the 2 registers. We used a longer time interval between the registrations in the general practitioner database, because women may have more and longer duration of follow-up visits with their general practitioners after a miscarriage. The routine postpartum checkup is done in general practice 6 weeks after delivery, and we required at least additional 6 weeks for the next registration to be defined as a new pregnancy. Second, we required that a record of a miscarriage or induced abortion in the patient or general practice databases be at least 6 weeks (patient registry) or 3 months (general practitioner database) after a registered delivery in the birth registry. Third, we excluded events registered as miscarriages or induced abortions that occurred within the gestational period of a registered pregnancy in the birth registry. For the 2 codes of threatened abortion (patient registry) and bleeding during pregnancy (general practitioner database), these were counted as miscarriages only if they did not have another code indicating the outcome of the pregnancy within 90 days.

## Maternal chronic diseases

Diagnoses of chronic diseases were obtained from the patient and general practitioner databases. We included autoimmune diseases (autoimmune thyroiditis, systemic lupus erythematosus, type 1 diabetes, celiac disease, multiple sclerosis, rheumatoid arthritis/ankylosing spondylitis, ulcerative colitis, psoriasis, Addison disease, Crohn's disease, haemolytic anemia), endocrinological diseases (Cushing syndrome, hypothyroidism, hyperthyroidism, hyperaldosteronism, hyperparathyroidism), cardiometabolic diseases (atherosclerosis, hypertensive disorders, type 2 diabetes), allergic diseases (asthma, allergic rhinitis, atopic dermatitis), neurological diseases (epilepsy, migraine), and reproductive diseases (polycystic ovary syndrome (PCOS) and endometriosis). We decided a priori what chronic diseases to include as exposures. The only notable changes based on what was decided a priori was the inclusion of an analysis of any cardiometabolic, any autoimmune, any reproductive, any endocrinological, any allergic, and any neurological diseases. In addition, autoimmune thyroiditis was included as part of review process. The various ICD-10 codes (specialist healthcare services) and ICPC-2 codes (primary healthcare services) used to capture the chronic disorders are listed in S1 Table. To reduce potential misclassification, we required 2 or more registrations of these

diagnostic codes, with at least 1 registration prior to the pregnancy of interest. We estimated the duration of pregnancies in the birth registry by subtracting the estimated gestational age in days from the date of the birth. Since we did not have information on the gestational age of miscarriages or induced abortions identified in the patient and general practitioner databases, we required that a diagnosis of chronic diseases for these pregnancies be at least 12 weeks prior. Pregnancies with longer gestation than 12 weeks should be recorded in the birth registry.

## Statistical analysis

We used generalized estimating equations with an exchangeable correlation structure to calculate the odds ratios (ORs) of miscarriage according to chronic conditions. We adjusted for the woman's age at the start of pregnancy as a linear and squared term. We conducted additional multivariable analyses adjusting for the number of previous pregnancies. We also explored multivariable analyses with mutual adjustment for all underlying chronic diseases in the same model.

The risk of miscarriage is typically defined as the number of miscarriages divided by the sum of miscarriages and births. This definition does not take into account the competing risk of induced abortions. Pregnant women who intend to terminate their pregnancy are at risk of miscarriage up to the time of termination, but (by definition) there is no risk of birth. Miscarriages among these pregnancies contribute to the numerator of miscarriage risk with no corresponding contribution to the denominator, thus inflating the risk estimate for miscarriage. This bias can be adjusted by adding a portion of induced abortions to the denominator of miscarriage risk. We included 20% of induced abortions in the comparison group to account for the competing risk due to induced abortions. Details on the calculation of this adjustment value are provided in the Supplement Methods (S1 Text).

We explored stratified analyses by age (<35 years and 35 years and older) and by registration of the miscarriage in the specialist versus primary healthcare services. We also conducted a sensitivity analysis excluding molar pregnancies from the definition of miscarriage.

All analyses were conducted using STATA version 15 (Statacorp, Texas).

## Patient and public involvement

No patients were involved in setting the research question or the outcome measures, nor were they involved in developing plans for recruitment, design, or implementation of the study. No patients were asked to advise on interpretation or writing up of results. There are no plans to disseminate the results of the research to study participants or the relevant patient community.

## Results

We identified 593,009 pregnancies to 278,159 women with an estimated date of conception between January 1, 2010 and December 31, 2016 (Fig 1). Of these pregnancies, 409,952 (69.1%) ended in a live birth, 1,740 (0.3%) in stillbirth, 95,641 (16.1%) in induced abortion, while 85,676 (14.5%) in miscarriage. Miscarriage risk as a proportion of miscarriages plus live and stillbirths was 17.3%, reduced to 16.6% after adjusting for the competing risk of induced abortions. The mean age of women at the start of pregnancy was 29.7 years (SD 5.6 years). Fewer than 18% of women had a recorded chronic illness before pregnancy (Table 1), with the most common condition being allergic rhinitis (4%).

After adjusting for women's age, the category of illness with the strongest relation to miscarriage risk was cardiometabolic diseases (OR 1.25; 95% CI: 1.20, 1.31; *p*-value <0.001) (Fig 2). All 3 conditions in this category had substantial associations with miscarriage risk:

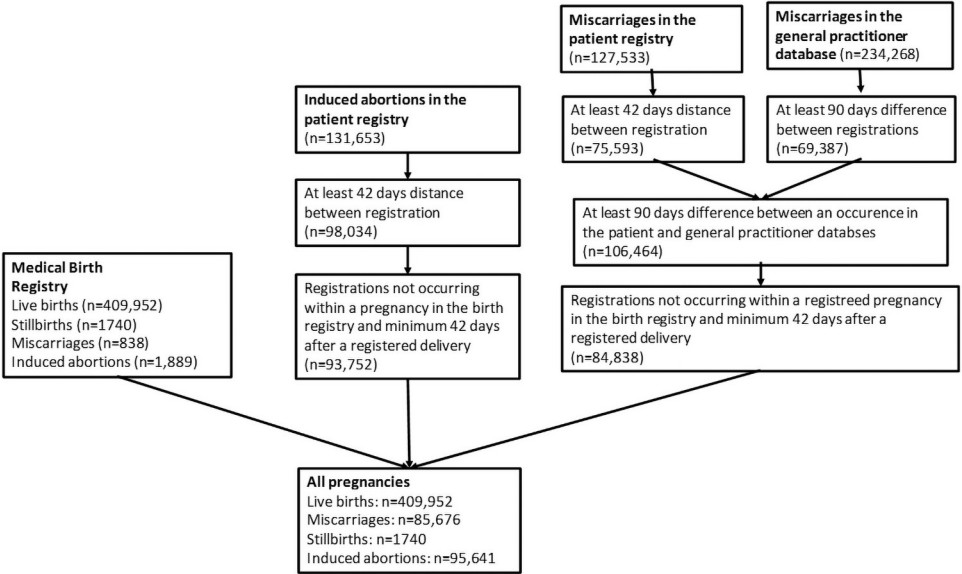

**Fig 1. Illustration of identification of unique pregnancies across the birth, patient, and general practitioner databases.**

**Table 1. Prevalence of chronic diseases prior to pregnancy among 593,009 pregnancies in Norway between 2010 and 2016.**

| Group of diseases | Diseases | No. pregnancies to women with underlying chronic diseases (% out of all pregnancies) | No. pregnancies to women with chronic diseases ending in a miscarriage (% out of exposed pregnancies) |
|---|---|---|---|
| Autoimmune diseases | Type 1 diabetes | 1,808 (0.30) | 289 (18.6) |
| | Celiac disease | 791 (0.13) | 130 (18.8) |
| | Systemic lupus erythematosus | 246 (0.04) | 43 (20.2) |
| | Multiple sclerosis | 720 (0.12) | 162 (19.4) |
| | Rheumatoid arthritis/ Ankylosing spondylitis | 2,559 (0.43) | 418 (18.7) |
| | Ulcerative colitis | 2,817 (0.48) | 456 (18.3) |
| | Psoriasis | 3,160 (0.53) | 601 (17.4) |
| | Crohn's disease | 1,603 (0.27) | 289 (21.0) |
| | Addison disease | 56 (0.01) | 11 (22.9) |
| | Haemolytic anemia | 194 (0.03) | 34 (21.1) |
| | Autoimmune thyroiditis | 455 (0.08) | 80 (20.9) |
| Cardiometabolic diseases | Type 2 diabetes | 1,951 (0.33) | 428 (25.1) |
| | Hypertensive disorders | 5,334 (0.90) | 1,085 (23.2) |
| | Atherosclerosis | 66 (0.01) | 21 (42.9) |
| Endocrinological diseases | Hypothyroidism | 8,372 (1.41) | 1,394 (18.6) |
| | Hyperthyroidism | 2,374 (0.40) | 378 (18.1) |
| | Hypoparathyroidism | 29 (0.005) | 9 (34.6) |
| | Hyperparathyroidism | 128 (0.02) | 22 (21.0) |
| | Cushing syndrome | 31 (0.01) | 10 (34.4) |
| Neurological diseases | Epilepsy | 2,563 (0.43) | 375 (17.8) |
| | Migraine | 18,594 (3.14) | 2,851 (17.7) |
| Allergic diseases | Asthma | 16,294 (2.75) | 2,405 (17.4) |
| | Allergic rhinitis | 23,652 (3.99) | 3,393 (16.2) |
| | Atopic dermatitis | 7,674 (1.29) | 1,020 (15.6) |
| Reproductive diseases | Polycystic ovary syndrome | 83 (0.01) | 20 (24.7) |
| | Endometriosis | 4,827 (0.81) | 1,011 (22.4) |

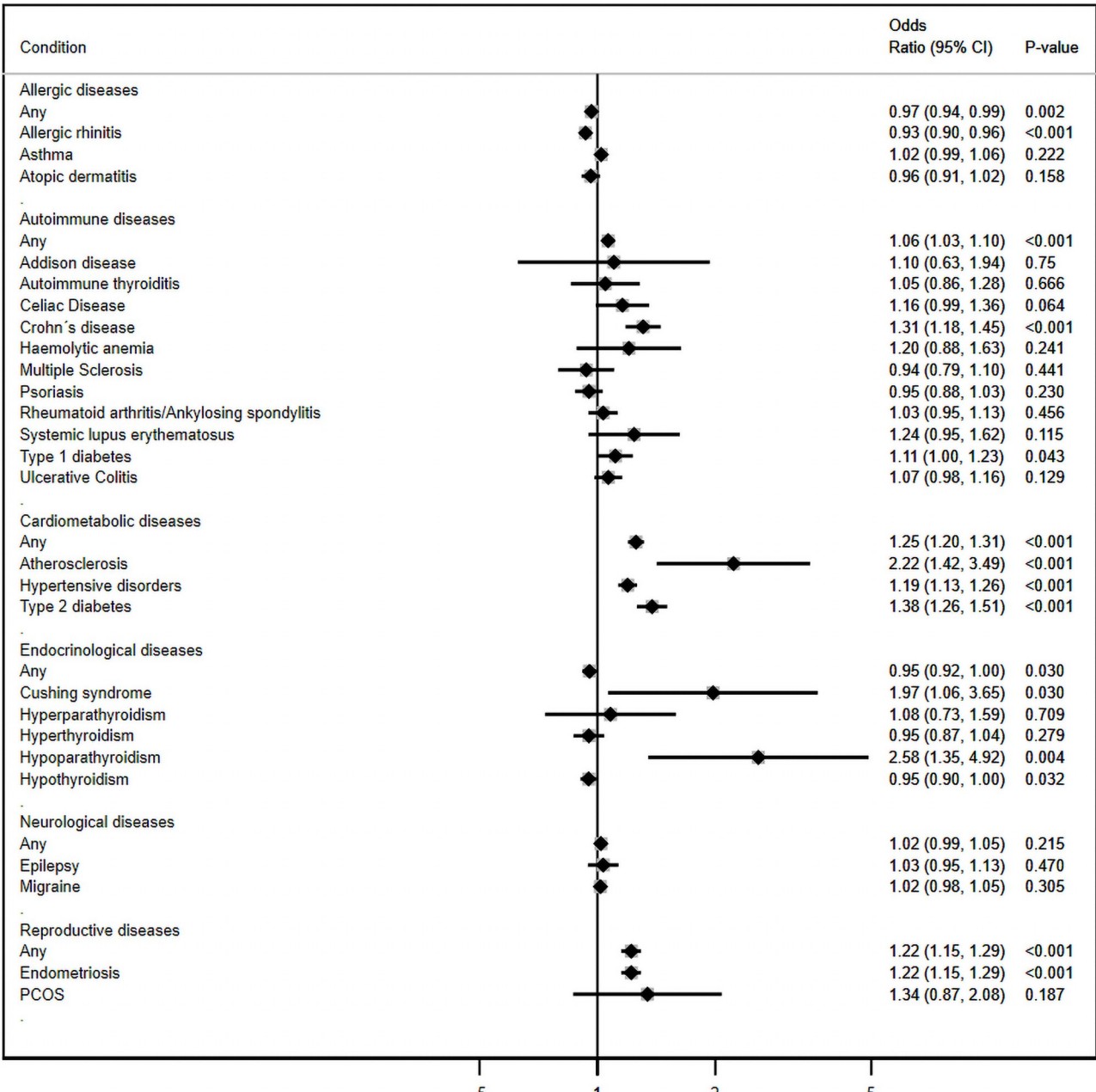

**Fig 2. Adjusted odds ratios (ORs)\* of miscarriage according to the presence of chronic conditions prior to the start of pregnancy (*n* = 593,009).**
\*The ORs are adjusted for the woman's age at the start of pregnancy as a linear and a squared term.

atherosclerosis (OR 2.22; 95% CI: 1.42 to 3.49; *p*-value <0.001), type 2 diabetes (OR 1.38; 95% CI: 1.26 to 1.51; *p*-value <0.001), and hypertensive disorders (OR 1.19; 95% CI: 1.13 to 1.26; *p*-value <0.001). The category of reproductive diseases was the only other category with substantial risk (OR 1.22; 95% CI: 1.15, 1.29; *p*-value <0.001), although there were only 2 conditions in this category (endometriosis (OR 1.22; 95% CI: 1.15 to 1.29; *p*-value <0.001) and PCOS (OR 1.34; 95% CI: 0.87 to 2.08; *p*-value 0.2).

Among the other categories, a few conditions showed strong associations, including hypoparathyroidism (OR 2.58; 95% CI: 1.35 to 4.92; *p*-value 0.004), Cushing syndrome (OR 1.97;

**Table 2. The adjusted\* odds ratios (OR) of miscarriage according to number of preexisting chronic conditions registered prior to pregnancy in 593,009 pregnancies.**

| Number of chronic diseases present before pregnancy | Adjusted\* OR | 95% CI |
|---|---|---|
| 0 | 1.00 (reference) | |
| 1 | 1.01 | 0.99, 1.04 |
| 2 | 1.14 | 1.07, 1.20 |
| 3 or more | 1.49 | 1.25, 1.78 |

\*Adjusted for the woman's age at the start of pregnancy.

95% CI: 1.06 to 3.65; *p*-value 0.03), Crohn's disease (OR 1.31; 95% CI: 1.18 to 1.45; *p*-value <0.001), and type 1 diabetes (OR 1.11; 95% CI: 1.01, 1.23; *p*-value 0.04). Adjustment for woman's age was important for unbiased estimates, as shown by the attenuation of the unadjusted results (S1 Fig). The sensitivity analysis excluding molar pregnancies from the definition of miscarriage showed similar results as the main analysis (S2 Fig).

Adjusted results were largely consistent when stratified by women's age (dichtomized at age 35) (S3 Fig), with some exceptions. Miscarriage risk was far stronger among older women with atherosclerosis and PCOS than among younger, while the risk of miscarriage associated with Cushing syndrome and hyperparathyroidism was stronger among the younger women.

Women with miscarriages who were seen only by general practitioners were slightly younger (mean 0.8 years) and less likely to have chronic disorders than women seen by specialists (S2 Table). The risk of miscarriage by maternal age was the same for miscarriages identified in the patient and general practitioner databases (S4 Fig). The associations between chronic conditions and risk of miscarriage were confined largely to miscarriages referred to specialist healthcare services (S5 Fig).

The risk of miscarriage was higher if a woman had more than one chronic disease. The OR of miscarriage with any one chronic disease was 1.01 (95% CI: 0.99 to 1.04; *p*-value 0.4), increasing to 1.14 (95% CI: 1.07 to 1.20; *p*-value <0.001) with any two and 1.49 (95% CI: 1.25 to 1.78; *p*-value <0.001) with any three (Table 2). The main findings were robust to mutual adjustment for other underlying chronic disorders (S6 Fig).

## Discussion

Combining information from 3 national health registries, we found that the risk of miscarriage was largely unaffected by the presence of a chronic disease in the mother. A distinct increase in miscarriage risk was present among women with cardiometabolic diseases and a few other illnesses (Crohn's disease, Cushing syndrome, hypoparathyroidism, and endometriosis). Women with two or more chronic diseases were at increased risk regardless of their illness.

It has previously been shown that women with a history of miscarriages are at increased risk of subsequent cardiovascular disease [33]. Our data suggest that miscarriage risk does not cause cardiovascular disease, but rather that preexisting cardiometabolic health might influence the risk of miscarriage. We observed a clear pattern of increased miscarriage risk across all 3 diseases included in our category of cardiometabolic diseases (atherosclerosis, hypertension, and type 2 diabetes). One previous study has reported a link between hypertension and miscarriage risk [34], although this has not been consistently found [35]. Previous studies also support an increased risk of miscarriage among women with type 2 diabetes, including 1 case–control study that retrospectively collected information on history of miscarriages, and 2 clinical chart reviews [9,19,20].

Taking these results together, cardiometabolic health seems an important contributor to a successful pregnancy. It is already well established that women with cardiovascular risk factors are at increased risk of preeclampsia and preterm delivery [36]. It would not be surprising if miscarriage risk should be added to this list. Women undergo extensive hemodynamic and metabolic changes during pregnancy, with an increase in blood volume, heart rate, glucose levels, and an altered lipid profile [37–39]. It is plausible that women with poorer cardiometabolic health prior to pregnancy (such as elevated blood pressure or glucose levels) might not adapt as well to pregnancy, rendering them at higher risk of miscarriage. These underlying biological mechanisms deserve further exploration. An increased risk of miscarriage among women with Cushing syndrome could also be related to cardiometabolic mechanisms [23].

A meta-analysis of 9 retrospective studies of women with endometriosis suggested an increased history of miscarriage [40]. A previous prospective study also found an increased risk with endometriosis [41]. A possible biological pathway might be the progesterone resistance in women with endometriosis, leading to dysregulation of genes important in embryo implantation and the subsequent greater risk of pregnancy loss [42,43].

A higher risk of miscarriage among women with Crohn's disease has not previously been reported, although other autoimmune diseases have been linked to miscarriage risk [7–11,23]. We likewise saw borderline increased risk with other autoimmune conditions. Pregnant women undergo a shift from a Th1 immune profile (which often characterizes autoimmune disorders) to a Th2 profile (which often characterizes allergic disorders), and this shift helps to ensure a successful pregnancy by facilitating an immune tolerance towards the embryo [44,45]. An insufficient immune adaptation during pregnancy may be associated with an increased risk of fetal loss [46].

## Limitations and strengths

Our study had important limitations. We acknowledge that the strict definition of miscarriage refers to uterine pregnancies. In our registry study, we defined the outcome as miscarriages or pregnancies that did not survive to 16 weeks; ectopic pregnancies were excluded. We were also unable to adjust for some potential confounding factors. Women's socioeconomic position and lifestyle characteristics (e.g., body mass index and smoking during pregnancy) have been reported to be associated with the risk of miscarriage [47–49]. Uncontrolled confounding by these factors might have contributed to our results, especially for hypertension and type 2 diabetes. We used E-values to estimate the magnitude of the relationship that any unmeasured confounder would have to have to completely explain away our findings [50]. An unmeasured confounder would have to be associated with a 2-fold increased risk of both miscarriage and type 2 diabetes to completely attenuate their observed association, while the unmeasured confounder would have to be associated with a 1.74 increased risk of both miscarriage and endometriosis to completely attenuate their observed association. We were able to identify only miscarriages that resulted in contact with healthcare services (primary or secondary healthcare services). Very early miscarriages where the woman did not know she was pregnant or did not choose to seek medical attention could not be included. We also acknowledge that defining pregnancy outcomes based on administrative codes in the registries might have caused misclassification due to our inability to confirm the outcome by clinical examinations. Our findings might also not be generalizable to populations with different ethnic compositions or populations without universal healthcare services. We might have underestimated the number of women with chronic diseases based on information from the registries. This could have resulted in an attenuation of the observed associations due to the presence of chronic diseases

also among some women defined as disease free. However, the registration of chronic diseases in the registries have been shown to have a high validity [31,51].

We also did not have information on treatment the women may have been receiving for these conditions, so we cannot distinguish between the effects of underlying diseases and the medications used to treat those diseases. The proportion of women with preexisting chronic conditions is also likely underestimated, as compared with the prevalence of some of these diseases from other pregnant populations [52,53]. This may be due in part to our conservative approach, requiring 2 diagnostic registrations in order to assign a disease to a woman.

In our previous study [32], based on pregnancies in the birth registry and the patient registry, the total risk of miscarriage risk was 13%, which captured miscarriages seen by specialists healthcare services. In the present analysis, we were able to add miscarriages diagnosed by general practitioners, which increased the risk of miscarriage to 16%. While this is at the upper end of the usual range, it is similar to reports from a British general practitioner database [54]. Our study is, to our knowledge, unique in its size (a whole nation), its inclusion of all recognized pregnancies seen by primary or specialist healthcare services, and its basis in recorded medical data.

In conclusion, maternal chronic diseases were largely unassociated with increased risk of miscarriage, although diseases related to cardiometabolic health were consistently linked to elevated miscarriage risk. A better understanding of the potential biological processes underlying these associations might eventually suggest ways to improve pregnancy outcomes.

## Supporting information

**S1 RECORD Checklist.**
(DOCX)

**S1 Analysis Plan.**
(DOCX)

**S1 Text. Supplementary methods.**
(DOCX)

**S1 Table. Diagnostic codes used to define chronic diseases in specialist care (ICD-10 codes), and in primary care (ICPC-2 codes).**
(DOCX)

**S2 Table. Prevalence of preexisting chronic diseases prior to pregnancy within miscarriages identified in the specialist and primary healthcare services.**
(DOCX)

**S1 Fig. Unadjusted odds ratios of miscarriage according to the presence of chronic conditions prior to pregnancy.**
(DOCX)

**S2 Fig. Adjusted odds ratios of miscarriage according to the presence of chronic conditions prior to pregnancy excluding molar pregnancies.**
(DOCX)

**S3 Fig. Adjusted odds ratios of miscarriage according to the presence of chronic conditions prior to pregnancy stratified by whether the mother was younger than 35 years ($n$ = 481,579) or 35 years or higher ($n$ = 111,430).**
(DOCX)

**S4 Fig. Age-associated risk of miscarriage identified in specialist (*n* = 62,974 pregnancies) or primary (*n* = 22,702 pregnancies) healthcare services.**
(DOCX)

**S5 Fig. Adjusted odds ratios of miscarriage according to the presence of chronic conditions prior to pregnancy according to whether the miscarriage was identified in the specialist (*n* = 62,974 pregnancies) or primary healthcare (*n* = 22,702 pregnancies).**
(DOCX)

**S6 Fig. Adjusted odds ratios of miscarriage according to the presence of chronic conditions prior to pregnancy mutually adjusting for other chronic conditions.**
(DOCX)

## Author Contributions

**Conceptualization:** Maria C. Magnus, Allen J. Wilcox, Siri E. Håberg.

**Formal analysis:** Maria C. Magnus.

**Funding acquisition:** Siri E. Håberg.

**Investigation:** Maria C. Magnus, Nils-Halvdan Morken, Knut-Arne Wensaas, Allen J. Wilcox, Siri E. Håberg.

**Methodology:** Maria C. Magnus, Nils-Halvdan Morken, Knut-Arne Wensaas, Allen J. Wilcox, Siri E. Håberg.

**Project administration:** Maria C. Magnus.

**Writing – original draft:** Maria C. Magnus.

**Writing – review & editing:** Nils-Halvdan Morken, Knut-Arne Wensaas, Allen J. Wilcox, Siri E. Håberg.

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
