## [Editor Report · Decision Letter 0]

17 Aug 2020

Dear Dr Magnus, 

Thank you for submitting your manuscript entitled "Risk of miscarriage in women with chronic diseases" for consideration by PLOS Medicine.

Your manuscript has now been evaluated by the PLOS Medicine editorial staff as well as by an academic editor with relevant expertise and I am writing to let you know that we would like to send your submission out for external peer review.

Kind regards,

Artur Arikainen,

Associate Editor

PLOS Medicine

---

## [Decision Letter · Decision Letter 1]

30 Nov 2020

Dear Dr. Magnus,

Thank you very much for submitting your manuscript "Risk of miscarriage in women with chronic diseases" (PMEDICINE-D-20-03922R1) for consideration at PLOS Medicine. 

Your paper was evaluated by a senior editor and discussed among all the editors here. It was also discussed with an academic editor with relevant expertise, and sent to three independent reviewers, including a statistical reviewer (r#1). The reviews are appended at the bottom of this email and any accompanying reviewer attachments can be seen via the link below:

[LINK]

In light of these reviews, I am afraid that we will not be able to accept the manuscript for publication in the journal in its current form, but we would like to consider a revised version that addresses the reviewers' and editors' comments. Obviously we cannot make any decision about publication until we have seen the revised manuscript and your response, and we plan to seek re-review by one or more of the reviewers. 

We expect to receive your revised manuscript by Dec 21 2020 11:59PM. Please email us (plosmedicine@plos.org) if you have any questions or concerns.

We look forward to receiving your revised manuscript. 

Sincerely,

Emma Veitch, PhD

PLOS Medicine

On behalf of Artur Arikainen, PhD, Associate Editor, 

PLOS Medicine

plosmedicine.org

*Please revise your title according to PLOS Medicine's style - we'd suggest this includes an indication of the study design (eg, "registry linkage study) in the subtitle (ie, after a colon).

*In the paper the authors define the study as a prospective-registry-based study, however given the analysis uses routinely collected data (the medical birth registry, the Norwegian Patient Registry, and the general practitioner database), presumably the data existed in those registries before the plan to draw them together and analyse them for the purposes of this question, was developed? In which case it might be better to define it as retrospective (in the sense the data existed before the analysis plan)? More clarity on this would be helpful.

*We'd suggest using the RECORD guideline (developed to help reporting of analyses conducted using registry linkage/routinely collected data) - https://www.equator-network.org/reporting-guidelines/record/. In this case please include the completed RECORD checklist as Supporting Information. Please add the following statement, or similar, to the Methods: "This study is reported as per the RECORD guideline (S1 Checklist)". When completing the checklist, please use section and paragraph numbers, rather than page numbers.

*In the last sentence of the Abstract Methods and Findings section, please include a brief note about any key limitation(s) of the study's methodology. Specifically, a reviewer notes that there are major limitations in terms of the ability to adjust for possible (likely) confounders in this study, and this certainly should be mentioned. 

*Can the authors clarify at what point the analytical plan used in this paper was developed - please state this (either way) early in the Methods section.

Comments from the reviewers:

Reviewer #1: I confine my remarks to statistical aspects of this paper. These were well done. My only concern is about confidence intervals (I am glad the authors did not include p values). But, since the authors have the entire population, what are these CIs? Some people posit a "super-population" of some sort. I'm not a big fan of this, pregnancies in Norway are not a random sample of pregnancies world wide (for one thing, health care is much better there than in almost any other country). But this needs to be addressed.

I'd be fine with no CIs, but others might object.

Peter Flom

Reviewer #2: This is a fairly straightforward analysis of whether chronic disease is associated miscarriage. Strengths of the paper are the large, population-based dataset, consistent coding, and careful consideration of analytic issues, such as competing risk due to induced abortion.

The authors discuss the possibility of missing diagnosis of chronic disease, but have less to say about missing diagnosis of miscarriage. Many early miscarriages are not clinically treated, many are probably not reported, and others are not even noticed. The authors discuss the overall incidence and how it compares to other populations, but not this issue. I also don't see discussion of the fact that women under treatment for some conditions may also be more aware of their bodies, or more likely to report to doctors if they do miscarry, particularly for a condition like endometriosis. An additional possible issue is competing risk from subfertility, which might be a cause for the lack of relationship with some outcomes (it is possible that some of the medications used to treat epilepsy could cause very early, i.e. undetectable, losses or infertility.) 

The authors do consider the possibility of confounding, which they are able to address only to a limited extent. I would like to see this emphasized a bit more in the abstract.

Reviewer #3: This study reports on the prospective risk of miscarriage in women with a preexisting chronic disease based on 6 years of all reported pregnancies in Norway (2010-2016). The study design ensures that the chronic disease is diagnosed prior to the pregnancy loss. The authors also undertake the task of estimating how many of the terminated pregnancies would have ended as a pregnancy loss if not the termination had been undertaken. The study is larger than previous studies, however no new information is given by the study.

Major issues. 

Terminology: Miscarriage is per definition a loss of a confirmed intrauterine pregnancy. This requires that the pregnancy has been scanned, identification of villi or fetus in the expelled pregnancy or in the evacuate. The authors use miscarriage despite several of their included pregnancies are not confirmed intrauterine - please instead use pregnancy loss. 

Definition: It is unclear why the authors included molar pregnancies in their definition of miscarriage. Please explain.

Selection of chronic diseases: On what basis was chronic diseases included in the analysis - why was Hashimoto thyroiditis not included? Known to be the most common autoimmune manifestation in women of reproductive age and previously found to be associated with the risk of pregnancy loss. 

Validation of diagnosis: Please include information of diagnosis validation of the included diagnoses. It is surprising that only 1% of the included women suffer from PCOS. This is the most common endocrine disorder in reproductive aged women and supposed to be as high as 5-10%. 

Why are numbers of pregnancy losses (the core of this manuscript) not mentioned in Table 1 as it is in supplementary table 2?

[LINK]

---

## [Decision Letter · Decision Letter 2]

19 Jan 2021

Dear Dr. Magnus,

Thank you very much for re-submitting your manuscript "Risk of miscarriage in women with chronic diseases: a Norwegian registry linkage study" (PMEDICINE-D-20-03922R2) for review by PLOS Medicine.

I have discussed the paper with my colleagues and the academic editor and it was also seen again by two reviewers. I am pleased to say that provided the remaining editorial and production issues are dealt with we are planning to accept the paper for publication in the journal.

[LINK]

We look forward to receiving the revised manuscript by Jan 26 2021 11:59PM.   

Sincerely,

Artur Arikainen

Associate Editor

PLOS Medicine

plosmedicine.org

Requests from Editors:

1. Please address the reviewer #3’s comment on correct use of terminology – this has also been requested by the Academic Editor. Further consideration of your manuscript is contingent on this being addressed.

2. Title: Please amend to: “Risk of pregnancy loss in women with chronic diseases in Norway: A registry linkage study”

3. Short title: Please amend to: “Chronic diseases and pregnancy loss”

4. Please include line numbers in the margin throughout.

5. Abstract: 

a. Please remove “prospectively” here and throughout, since your analysis itself is retrospective.

b. Please remove “retrospective” when describing the study here and throughout, since “registry-based” already implies a retrospective analysis.

c. Please include p values for your comparisons, alongside 95% CIs.

d. At the end of the ‘Methods and findings’ subsection, please state more clearly: “Limitations of this study were…”. You could also mention the rareness of some conditions as a further limitation, eg. Cushing’s syndrome.

e. Conclusions: Please begin with “In this registry study, we found that…”

6. Page 3: Delete keywords.

7. Author Summary: Amend to: “To our knowledge, none of the existing studies have…”

8. Please remove spaces from within citations, eg: “…environmental factors [5,6].”

9. Results: Please include p values for your comparisons, alongside 95% CIs.

10. Discussion: Please amend to: “Our study is, to our knowledge, unique in its…”

11. Pages 17-18: Please remove the Author Contributions, Data Availability Statement, Funding, and Competing interests sections – these are taken from the online submission form.

12. References:

a. Please remove city and country from journal names, eg. “(London, England)”.

b. Please delete this from reference 32: “at www.icmje.org/coi_disclosure.pdf and declare: no support from any organisation for the submitted work; no financial relationships with any organisations that might have an interest in the submitted work in the previous three years; no other relationships or activities that could appear to have influenced the submitted work.”

c. Please delete this from reference 36: “Copyright 2018, The Author(s).”

13. Please add the following statement, or similar, to the Methods: "This study is reported as per the Strengthening the Reporting of Observational Studies in Epidemiology (STROBE) and REporting of studies Conducted using Observational Routinely-collected Data (RECORD) guidelines (S1 Checklist)." (and rename the file accordingly)

14. When completing the STROBE/RECORD checklist, please use section and paragraph numbers, rather than page numbers.

---

Comments from Reviewers:

Reviewer #1: The authors have addressed my concerns and I now recommend publication

Peter Flom

Reviewer #3: I am highly surprised that this experienced and respected author group in their rebuttal to PLOS MEDICINE have not responded to a reviewer comment and therefore without any notice keep using miscarriage. To use miscarriage, they should be certain that the pregnancy had been confirmed intrauterine. In practical terms had been scanned, or villi had been identified or the fetus had been visible in the bleeding or the pregnancy material. Clinical researchers changed the terminology to pregnancy loss several years ago unless certain that the losses are intrauterine. I think women pregnant with a gestational age <12 and no prior ultrasound scans with a large bleeding prior to or on their way to hospital who present with an empty uterus at ultrasound and no evaluation of the bleeding will be included in this study.

[LINK]

---

## [Editor Report · Decision Letter 3]

27 Jan 2021

Dear Dr. Magnus,

Thank you very much for re-submitting your manuscript "Risk of miscarriage in women with chronic diseases: a Norwegian registry linkage study" (PMEDICINE-D-20-03922R3) for review by PLOS Medicine.

I am pleased to say that provided the remaining editorial and production issues are dealt with we are planning to accept the paper for publication in the journal.

The remaining minor issues that need to be addressed are listed at the end of this email.

We look forward to receiving the revised manuscript by Feb 03 2021 11:59PM.   

Sincerely,

Artur Arikainen, 

Associate Editor 

PLOS Medicine

plosmedicine.org

Requests from Editors:

1. Title: Thank you for your reply regarding the terminology – this is fine to leave as is, as we discussed. However, please nevertheless amend to the following to better fit our journal style: “Risk of miscarriage in women with chronic diseases in Norway: A registry linkage study”

2. Abstract: Please add some brief summary demographics for participants (eg. age).

3. Results/Figures and Abstract: Please present p values to 3 decimal places consistently, or “p<0.001”.

4. When completing the STROBE/RECORD checklist, please use section and paragraph numbers, rather than line or page numbers (these will change in the final published version).

Comments from Reviewers:

n/a

[LINK]

---

## [Editor Report · Decision Letter 4]

28 Mar 2021

Dear Dr. Magnus,

Thank you very much for re-submitting your manuscript "Risk of miscarriage in women with chronic diseases in Norway: A registry linkage study" (PMEDICINE-D-20-03922R4) for consideration at PLOS Medicine. We apologize for the delay in contacting you: Dr Arikainen has recently left the journal. 

I have discussed the paper with editorial colleagues and I am pleased to tell you that, provided the remaining editorial and production issues are fully dealt with, we expect to be able to accept the paper for publication in the journal.

The remaining issues that need to be addressed are listed at the end of this email. Please take these into account before resubmitting your manuscript.

Please let me know if you have any questions, and we look forward to receiving the revised manuscript shortly.   

Sincerely,

Richard Turner PhD

rturner@plos.org

Requests from Editors:

Please amend your data statement to: "Study data are available on application via helsedata.no, subject to the necessary ethics approvals." or similar.

Please make that "risk of miscarriage" at line 57.

Please state early in the Methods section whether the study had a protocol or prespecified analysis plan. Please highlight analyses that were not prespecified. 

Our Academic Editor has requested that you deal with referee 3's comment explicitly in your discussion section. We ask that you do this by amending the wording at line 322, and suggest: "Our study had important limitations. We acknowledge that the strict definition of miscarriage refers to uterine pregnancies. In our registry study, we defined the outcome as miscarriages or pregnancies that did not survive to 16 weeks; ectopic pregnancies were excluded." or similar. 

***

---

## [Editor Report · Decision Letter 5]

1 Apr 2021

Dear Dr Magnus, 

On behalf of my colleagues and our Academic Editor Dr Myers, I am pleased to inform you that we have agreed to publish your manuscript "Risk of miscarriage in women with chronic diseases in Norway: A registry linkage study" (PMEDICINE-D-20-03922R5) in PLOS Medicine.

We ask you to address two further issues:

If available, please attach your study analysis plan as a supplementary document, referred to around line 106;

There is a sense of imbalance in the current presentation: around lines 50 and 66, for example, there is an emphasis on associations of miscarriage with cardiometabolic conditions, whereas in the Discussion (around lines 280 and 368) the emphasis is more on an absence of associations with chronic conditions, and we suggest amending the wording to harmonise the different parts of your paper. 

Before your manuscript can be formally accepted you will also need to complete some formatting changes, which you will receive in a follow up email. Please be aware that it may take several days for you to receive this email; during this time no action is required by you. Once you have received these formatting requests, please note that your manuscript will not be scheduled for publication until you have made the required changes.

PRESS

Sincerely, 

Richard Turner, PhD 

rturner@plos.org